# Fatigue-Induced Failure of Polysilicon MEMS: Nonlinear Reduced-Order Modeling and Geometry Optimization of On-Chip Testing Device

**DOI:** 10.3390/mi15121480

**Published:** 2024-12-08

**Authors:** Daniel Calegaro, Massimiliano Merli, Giacomo Ferrari, Stefano Mariani

**Affiliations:** 1Department of Civil and Environmental Engineering, Politecnico di Milano, Piazza Leonardo da Vinci, 32, 20133 Milano, Italy; stefano.mariani@polimi.it; 2STMicroelectronics, 20007 Cornaredo, Italy; massimiliano.merli@st.com (M.M.); giacomo.ferrari@st.com (G.F.)

**Keywords:** polysilicon MEMS, on-chip testing, reliability, fatigue-induced failure, reduced-order modeling, DPIM

## Abstract

In the case of repeated loadings, the reliability of inertial microelectromechanical systems (MEMS) can be linked to failure processes occurring within the movable structure or at the anchors. In this work, possible debonding mechanisms taking place at the interface between the polycrystalline silicon film constituting the movable part of the device and the silicon dioxide at the anchor points are considered. In dealing with cyclic loadings possibly inducing fatigue failure, a strategy is proposed to optimize the geometry of an on-chip testing device designed to characterize the strength of the aforementioned interface. Dynamic analyses are carried out to assess the deformation mode of the device and maximize the stress field leading to interface debonding. To cope with the computational costs of numerical simulations within the structural optimization framework, a reduced-order modeling procedure for nonlinear systems is discussed, based on the direct parametrization of invariant manifolds (DPIM). The results are reported in terms of maximum stress intensification for varying geometry of the testing device and actuation frequency to demonstrate the accuracy and computational efficiency of the proposed methodology.

## 1. Introduction

Microelectromechanical systems (MEMS) have revolutionized a wide range of applications by combining miniaturized mechanical and electrical components through advanced microfabrication techniques [1]. Their compact size, multi-functional capabilities, and scalability have driven their widespread adoption in fields such as aerospace, biomedical devices, and consumer electronics [2]. However, the reliability of MEMS in the presence of harsh environmental conditions remains a significant challenge, particularly when devices are subjected to high-cycle mechanical loadings, elevated temperature, or humidity [3]. The continuous evolution of MEMS hinges on addressing these reliability issues, particularly through a better understanding of the materials and failure mechanisms at play.

At the core of MEMS reliability concerns are two intertwined phenomena: fatigue and delamination. Fatigue failure, although not typically observed under standard operating conditions, can become a significant issue when MEMS are exposed to high stress levels. In parallel, delamination often occurs at critical interfaces, such as between silicon dioxide (SiO_2_) and polycrystalline silicon (polysilicon), where high interfacial stress develops under cyclic loadings. Together, these mechanisms progressively alter the mechanical properties of MEMS devices, leading to shifts in, e.g., resonance frequency and electrical resistance, ultimately compromising performance and reliability [4,5].

Addressing fatigue-related failures in MEMS requires testing capabilities at high frequencies—ideally above 1 kHz—so that millions of cycles can be achieved in a relatively short time. Traditional experimental setups often struggle to replicate these conditions, but on-chip test platforms offer a solution, enabling the assessment of fatigue and delamination phenomena under realistic operating conditions in a reasonable amount of time [6]. High-frequency testing is critical to understanding the subtle shifts in material behavior that lead to long-term failure.

This research study focuses on the optimization of the stress distribution in a MEMS testing device through a purpose-built test setup that incorporates piezoelectric actuation and sensing in a closed-loop configuration. The final goal is the amplification of the stress concentration at the SiO_2_-polysilicon interface, where fatigue-induced delamination is most likely to occur. In [7], we already took advantage of stationary finite element (FE) analyses to infer the best geometry to induce the highest stress at that interface. The results showed that it is possible to reach a geometric configuration at which the stress at the interface is maximum and larger than in the polysilicon film to increase the probability of a localized fracture at the interface itself, instead of the cracking of the movable structure.

However, in order to effectively replicate real working conditions, sinusoidal time-varying loadings must be taken into account, in place of static loads. The actuation frequency of a piezoelectric MEMS is ideally set close to the resonance frequency, giving rise to large amplitudes of oscillations and, consequently, structural nonlinearities. In this framework, the computation of the nonlinear dynamic responses becomes challenging due to the relevant burden. More specifically, where three-dimensional FE models are used to discretize accurately the device geometry and resolve the stress field at the SiO_2_-polysilicon interface, model-order reduction techniques become necessary to overcome these issues [8].

Model-order reduction techniques can be divided into linear and nonlinear ones. Linear methods, like modal projection, rely on a linear change in coordinates. In the presence of nonlinearities, the inefficiency of these methods quickly grows as the subspace dimension to obtain accurate solution increases, due to the loss of invariance of linear subspaces [9]. Conversely, nonlinear techniques rely on the extension of the properties of the linear modal basis to the nonlinear regime by preserving the invariance property [10]. These approaches bring several advantages compared with linear approaches, as they allow for the definition of a smaller subspace, due to enhanced efficiency, and the achievement of better accuracy in the solution.

Among the nonlinear reduction techniques, the direct parametrization of invariant manifolds (DPIM) looks promising, since it can be directly applied to systems expressed in the physical space, and not in the modal space, avoiding the definition of the eigenproperties, which may be computationally expensive in real structures with a high number of degrees of freedom [9]. This methodology is also more efficient compared with the technique employed in [11], where the invariant manifold was constructed after transitioning from the physical space to the phase space, thereby doubling the dimensionality of the system.

In this paper, the effects inducing fatigue delamination at the SiO_2_-polysilicon interface are studied for a MEMS testing device, piezoelectrically actuated with sinusoidal time-varying loadings. By maximizing the stress distribution at the aforementioned interface, the goal of a foreseen experimental campaign to be conducted in the future under varying environmental conditions is the triggering of decohesion. A numerical approach is here proposed to exploit the DPIM method and develop reduced-order models (ROMs) for forced-damped nonlinear dynamical systems. These ROMs are then integrated into an optimization strategy to assess the stress field in the structure under varying geometry and actuation conditions, aiming to maximize the stress state and possibly lead to a local failure.

## 2. Reduced-Order Modeling

Let the nonlinear vibrations of a mechanical system subject to periodic excitation be the focus of the analysis. Upon the space discretization of the governing equations through FEs, the equations of motion take the following form [12]:(1)$$M\ddot{U}+C\dot{U}+KU+G\left(U\right)=\varepsilon F_{0}\cos\left(\Omega t\right),\;0\leq \varepsilon \ll 1.$$

Here, the following apply: U is the time-dependent displacement vector of the system, representing the *N* degrees of freedom of the discretized model; *t* is time; a superposed dot denotes a time derivative, so that $\dot{U}$ and $\ddot{U}$ are, respectively, the velocity and acceleration vectors; M, C, and K $\in R^{N\times N}$ are the mass, damping, and stiffness matrices, respectively; the nonlinear internal force vector $G\left(U\right)=\sum_{k=2}^{3}G_{k}U^{⊗k}\in R^{N}$ is here assumed to consist of terms that are quadratic and cubic in U. $F_{0}\in R^{N}$ is the amplitude of the external forces; ε is a small scaling parameter; Ω is the angular excitation frequency. In the case ε=0, the system is autonomous; if, instead, ε>0, the system becomes non-autonomous [11]. As far as the specific notation here adopted is concerned, ⊗ denotes the Kronecker product, and $U^{⊗k}$ represents U⊗…⊗U (*k* times), containing $N^{k}$ monomial terms of degree *k* in U. For two generic matrices $A\in R^{i\times j}$ and $B\in R^{h\times k}$, the Kronecker product A⊗B gives as output the block matrix $D\in R^{ih\times jk}$; see [13]:(2)$$D=\left[\begin{array}{ccc} A_{11}B & \cdots & A_{1j}B\\ \vdots & \ddots & \vdots \\ A_{i1}B & \cdots & A_{ij}B \end{array}\right].$$

The damping term in Equation (Equation 1) is assumed to be of the Rayleigh type, so that the damping matrix C is given by a linear combination of the mass and stiffness matrices according to:(3)C=αM+βK,
where α and β are non-negative scalar coefficients, set by matching at best the experimentally measured Q factor(s); additional details can be found in Section 3.

### 2.1. Spectral Properties

The main spectral properties of the linearized system in the autonomous limit are first discussed. Moving from:(4)$$M\ddot{U}+C\dot{U}+KU=0,$$
and knowing that C is a linear combination of M and K, the relevant real eigenfunctions and eigenfrequencies are obtained by solving the problem:(5)$$\left(\lambda_{j}^{2}M+\lambda_{j}C+K\right)\phi_{j}=0,$$
with j=1,⋯,N. The orthogonality relations provide:(6)$$\phi_{i}^{T}M\phi_{j}=\delta_{ij},\;\phi_{i}^{T}C\phi_{j}=2\xi_{j}\omega_{j}\delta_{ij},\;\phi_{i}^{T}K\phi_{j}=\omega_{i}^{2}\delta_{ij},$$
where $\xi_{j}$ is the modal damping ratio.

For mechanical systems properly constrained to avoid rigid-body motion, both M and K are symmetric and positive definite matrices. Therefore, eigenfrequencies $\omega_{j}$ and eigenmodes $\phi_{j}$ are all real-valued. Eigenvalues $\lambda_{j}^{\pm }$ can then be obtained by projecting Equation (Equation 4) onto the eigenspace defined by the modal basis $\phi_{j}$:(7)$$\lambda_{j}^{\pm }=-\xi_{j}\omega_{j}\pm i\omega_{j}\sqrt{1-\xi_{j}^{2}}.$$
where $\lambda_{j}^{+}$ and $\lambda_{j}^{-}$ represent the solutions featuring the positive and negative signs, respectively, and i is the imaginary unit.

Eigenmodes $\phi_{j}$ and eigenvalues $\lambda_{j}$ form the foundation of the DPIM method. The model-order reduction technique of this method aims at approximating the high-dimensional solution to Equation (Equation 1) within a reduced-order or *master* subspace of order m≪N. Such a subspace represents a nonlinear extension of the linear eigenspace defined by the *m* selected eigenmodes, termed master modes, $\phi_{j}$, given by Equation (Equation 5).

The parametrization procedure requires the spectral properties of the master subspace. By considering the complex conjugate solutions of Equation (Equation 7), the spectral quantities related to the *m* selected master modes are arranged in the following matrices:(8)$$\begin{array}{cc} \; & \Phi =\left[\phi_{1},\phi_{2},\cdots ,\phi_{m}\right],\\ \; & \Lambda =\mathrm{diag}\left(\lambda_{1}^{+},\lambda_{2}^{+},\cdots ,\lambda_{m}^{+},\lambda_{1}^{-},\lambda_{2}^{-},\cdots ,\lambda_{m}^{-}\right). \end{array}$$

In the following, $\Phi_{j}$ refers to the *j*-th column of Φ and $\Lambda_{j}$ to the *j*-th diagonal element of Λ.

The DPIM procedure consists of different steps that are briefly outlined before going into a detailed description in the following sections. First, the time-invariant equation is derived by using a Taylor series expansion of the solution U in terms of a set of normal coordinates $p\left(t\right)\in C^{n}$ and of the parameter ε. Similarly, the reduced velocity $\dot{p}$ is expanded in terms of p and ε. Next, the coefficients arising from the aforementioned parametrization are determined. Finally, since U must be real-valued, the complex-valued nonlinear mapping and reduced velocity coefficients are transformed into real-valued ones.

### 2.2. Time-Invariant Equation

Dimensionality reduction is said to be accomplished by taking advantage of the Taylor series expansion of the solution vector U in terms of p and ε, i.e.:(9)$$U≃W_{\varepsilon }\left(p,\Omega t\right)=W_{0}\left(p\right)+\varepsilon W_{1}\left(p,\Omega t\right).$$

The reduced velocity $\dot{p}$ is accordingly defined as:(10)$$\dot{p}≃R_{\varepsilon }\left(p,\Omega t\right)=R_{0}\left(p\right)+\varepsilon R_{1}\left(p,\Omega t\right).$$

The time-invariant equation is then obtained by substituting Equations (Equation 9) and (Equation 10) into Equation (Equation 1) and by expressing $W_{\varepsilon }\left(p,\Omega t\right)$ and $R_{\varepsilon }\left(p,\Omega t\right)$ as polynomial expansions of arbitrary orders:(11)$$W_{\varepsilon }\left(p,\Omega t\right)=\sum_{i=1}^{q_{0}}W_{i,0}p^{⊗i}+\varepsilon \sum_{i=0}^{q_{1}}\left(W_{i,1}^{+}e^{i\Omega t}+W_{i,1}^{-}e^{-i\Omega t}\right)p^{⊗i},$$
(12)$$R_{\varepsilon }\left(p,\Omega t\right)=\sum_{i=1}^{q_{0}}R_{i,0}p^{⊗i}+\varepsilon \sum_{i=0}^{q_{1}}\left(R_{i,1}^{+}e^{i\Omega t}+R_{i,1}^{-}e^{-i\Omega t}\right)p^{⊗i},$$
where $q_{0}$ and $q_{1}$ are the orders of the expansions in the autonomous and non-autonomous cases, respectively; $W_{i,0}\in C^{N\times n^{i}}$ and $R_{i,0}\in C^{n\times n^{i}}$ are the *i*-th nonlinear mapping and reduced velocity coefficients in the autonomous case; $W_{i,1}^{+}$ and $W_{i,1}^{-}$ $\in C^{N\times n^{i}}$ are the *i*-th Fourier expansion coefficients in the non-autonomous case for the nonlinear mapping; and $R_{i,1}^{+}$ and $R_{i,1}^{-}$ $\in C^{n\times n^{i}}$ are the *i*-th Fourier expansion coefficients in the non-autonomous case for the nonlinear reduced velocity. Here and in what follows, ${\left(•\right)}^{+}$ and ${\left(•\right)}^{-}$ represent the quantities referring to $e^{i\Omega t}$ and $e^{-i\Omega t}$, respectively.

By combining Equations (Equation 1) and (Equation 9)–(Equation 12) and by separating the terms related to different orders of ε, the following time-invariant expressions are obtained:(13)$$\begin{array}{cc} \; & \mathrm{Order}\;\varepsilon^{0}\\ \; & M\left(\sum_{i=2}^{q_{0}}W_{i,0}R_{i}^{\left(\ddot{U},2,1\right)}p^{⊗i}+2\sum_{i=2}^{q_{0}}W_{i,0}R_{i}^{\left(\ddot{U},2,2\right)}p^{⊗i}\right)+C\sum_{i=2}^{q_{0}}W_{i,0}R_{i}^{\left(\ddot{U},1,1\right)}p^{⊗i}+\\ \; & +K\sum_{i=1}^{q_{0}}W_{i,0}p^{⊗i}=-[M(\sum_{i=1}^{q_{0}}W_{1,0}R_{1,0}R_{i,0}p^{⊗i}+\sum_{i=2}^{q_{0}}W_{1,0}R_{i,0}R_{i}^{\left(\ddot{U},1,1\right)}p^{⊗i}+\\ \; & +\sum_{i=3}^{q_{0}}\Gamma_{i,0}^{\ddot{U}}p^{⊗i})+C\left(\sum_{i=1}^{q_{0}}W_{1,0}R_{i,0}p^{⊗i}+\sum_{i=3}^{q_{0}}\Gamma_{i,0}^{\dot{U}}p^{⊗i}\right)+G_{2}\sum_{i=2}^{q_{0}}\Gamma_{i,0}^{G_{2}}p^{⊗i}+G_{3}\sum_{i=3}^{q_{0}}\Gamma_{i,0}^{G_{3}}p^{⊗i}], \end{array}$$
(14)$$\begin{array}{cc} \; & \mathrm{Order}\;\varepsilon^{1}\\ \; & M[-\Omega^{2}\sum_{i=0}^{q_{1}}\left(W_{i,1}^{+}e^{i\Omega t}+W_{i,1}^{-}e^{-i\Omega t}\right)p^{⊗i}+2i\Omega \sum_{i=1}^{q_{1}}\left(W_{i,1}^{+}e^{i\Omega t}-W_{i,1}^{-}e^{-i\Omega t}\right)R_{i}^{\left(\ddot{U},1,1\right)}p^{⊗i}+\\ \; & +\sum_{i=1}^{q_{1}}\left(W_{i,1}^{+}e^{i\Omega t}+W_{i,1}^{-}e^{-i\Omega t}\right)R_{i}^{\left(\ddot{U},2,1\right)}p^{⊗i}+2\sum_{i=2}^{q_{1}}\left(W_{i,1}^{+}e^{i\Omega t}+W_{i,1}^{-}e^{-i\Omega t}\right)R_{i}^{\left(\ddot{U},2,2\right)}p^{⊗i}]+\\ \; & +C\left[i\Omega \sum_{i=0}^{q_{1}}\left(W_{i,1}^{+}e^{i\Omega t}-W_{i,1}^{-}e^{-i\Omega t}\right)p^{⊗i}+\sum_{i=1}^{q_{1}}\left(W_{i,1}^{+}e^{i\Omega t}+W_{i,1}^{-}e^{-i\Omega t}\right)R_{i}^{\left(\ddot{U},1,1\right)}p^{⊗i}\right]+\\ \; & +K\sum_{i=0}^{q_{1}}\left(W_{i,1}^{+}e^{i\Omega t}+W_{i,1}^{-}e^{-i\Omega t}\right)p^{⊗i}=-\{M[i\Omega \sum_{i=0}^{q_{1}}W_{1,0}\left(R_{i,1}^{+}e^{i\Omega t}-R_{i,1}^{-}e^{-i\Omega t}\right)p^{⊗i}+\\ \; & +\sum_{i=0}^{q_{1}}W_{1,0}R_{1,0}\left(R_{i,1}^{+}e^{i\Omega t}+R_{i,1}^{-}e^{-i\Omega t}\right)p^{⊗i}+\sum_{i=1}^{q_{1}}W_{1,0}\left(R_{i,1}^{+}e^{i\Omega t}+R_{i,1}^{-}e^{-i\Omega t}\right)R_{i}^{\left(\ddot{U},1,1\right)}p^{⊗i}+\\ \; & +\sum_{i=1}^{q_{1}}\Gamma_{i,1}^{\ddot{U}}p^{⊗i}]+C\left[\sum_{i=0}^{q_{1}}W_{1,0}\left(R_{i,1}^{+}e^{i\Omega t}+R_{i,1}^{-}e^{-i\Omega t}\right)p^{⊗i}+\sum_{i=1}^{q_{1}}\Gamma_{i,1}^{\dot{U}}p^{⊗i}\right]+\\ \; & +G_{2}\sum_{i=1}^{q_{1}}\Gamma_{i,1}^{G_{2}}p^{⊗i}+G_{3}\sum_{i=2}^{q_{1}}\Gamma_{i,1}^{G_{3}}p^{⊗i}\}+\frac{1}{2}F_{0}\left(e^{i\Omega t}+e^{-i\Omega t}\right), \end{array}$$
where:(15)$$R_{i}^{\left(\ddot{U},1,1\right)}=\sum_{l=0}^{i-1}\left(I_{n}^{⊗l}⊗R_{1,0}⊗I_{n}^{⊗i-l-1}\right),$$
(16)$$R_{i}^{\left(\ddot{U},2,1\right)}=\sum_{l=0}^{i-1}\left(I_{n}^{⊗l}⊗R_{1,0}R_{1,0}⊗I_{n}^{⊗i-l-1}\right),$$
(17)$$R_{i}^{\left(\ddot{U},2,2\right)}=\sum_{k=0}^{i-2}\sum_{l=0}^{i-k-2}\left(I_{n}^{⊗l}⊗R_{1,0}⊗I_{n}^{⊗k}⊗R_{1,0}⊗I_{n}^{⊗i-k-l-2}\right).$$

The terms $\Gamma_{i,0}^{\dot{U}}$, $\Gamma_{i,0}^{\ddot{U}}$, $\Gamma_{i,0}^{G_{2}}$, $\Gamma_{i,0}^{G_{3}}$, $\Gamma_{i,1}^{\dot{U}}$, $\Gamma_{i,1}^{\ddot{U}}$, $\Gamma_{i,1}^{G_{2}}$, and $\Gamma_{i,1}^{G_{3}}$, are of a degree lower than *i* and for this reason placed on the right-hand side of the equations; see [11]. Their expressions are reported in Appendix A. $I_{n}\in R^{n\times n}$ is the identity matrix. The coefficients of the nonlinear mapping ($W_{i,0}$, $W_{i,1}^{+}$, and $W_{i,1}^{-}$) and nonlinear dynamics ($R_{i,0}$, $R_{i,1}^{+}$, and $R_{i,1}^{-}$) are found by separating the terms with different p orders and by solving iteratively the resulting homological equations. These equations and the relevant solutions are discussed in the following in terms of the coefficients at each order *i* for both the autonomous case ($\varepsilon^{0}$) and the non-autonomous case ($\varepsilon^{1}$).

### 2.3. Autonomous Case

At the leading order in p, namely, for i=1, Equation (Equation 13) becomes:(18)$$MW_{1,0}R_{1,0}R_{1,0}+CW_{1,0}R_{1,0}+KW_{1,0}=0,$$
which is solved through vectorization [11,13] as:(19)$$\left(R_{1,0}^{T}R_{1,0}^{T}⊗M+R_{1,0}^{T}⊗C+I_{n}⊗K\right)\mathrm{vec}\left(W_{1,0}\right)=0,$$
where $\mathrm{vec}\left(W_{1,0}\right)$ means that the matrix $W_{1,0}$ is transformed into a column vector by stacking its columns one below the others. In concrete terms, it is recalled that $R_{1,0}$ and $W_{1,0}$ collectively gather the sought solutions.

By choosing a diagonal form for $R_{1,0}$ and projecting Equation (Equation 19) onto the subspace spanned by master eigenmodes $\Phi_{j}$, one obtains:(20)$${\left(e_{l}⊗\Phi_{j}\right)}^{T}\left(R_{1,0}^{T}R_{1,0}^{T}⊗M+R_{1,0}^{T}⊗C+I_{n}⊗K\right)\mathrm{vec}\left(W_{1,0}\right)=0,$$
(21)$$\left(e_{l}^{T}⊗\Phi_{j}^{T}M\right)\left(I_{n}⊗I_{N}\right)\left(\lambda_{l}^{2}+2\chi_{j}\omega_{j}\lambda_{l}+\omega_{j}^{2}\right)\mathrm{vec}\left(W_{1,0}\right)=0,$$
where $\lambda_{l}$ is the *l*-th diagonal coefficient in $R_{1,0}$. It must be noted that the solution in terms of $\lambda_{l}$ is the same obtained by solving in Equation (Equation 5). Hence, $W_{1,0}$ and $R_{1,0}$ can be written as:(22)$$W_{1,0}=\left[\Phi ,\Phi \right],\;R_{1,0}=\Lambda .$$

For i≥2, Equation (Equation 13) yields:(23)$$\begin{array}{cc} M & \left(W_{i,0}R_{i}^{\left(\ddot{U},2,1\right)}+2W_{i,0}R_{i}^{\left(\ddot{U},2,2\right)}\right)+CW_{i,0}R_{i}^{\left(\ddot{U},1,1\right)}+KW_{i,0}=\\ = & -\left[M\left(W_{1,0}R_{1,0}R_{i,0}+W_{1,0}R_{i,0}R_{i}^{\left(\ddot{U},1,1\right)}\right)+CW_{1,0}R_{i,0}\right]+\\ \; & -\left(M\Gamma_{i,0}^{\ddot{U}}+C\Gamma_{i,0}^{\dot{U}}+G_{2}\Gamma_{i,0}^{G_{2}}+G_{3}\Gamma_{i,0}^{G_{3}}\right), \end{array}$$
which, in a way similar to i=1, is solved through vectorization:(24)$$\begin{array}{cc} {}[( & R_{i}^{(\ddot{U},2,1)T}+2R_{i}^{(\ddot{U},2,2)T})⊗M+R_{i}^{(\ddot{U},1,1)T}⊗C+I_{n^{i}}⊗K]\mathrm{vec}\left(W_{i,0}\right)=\\ = & -\left(I_{n^{i}}⊗MW_{1,0}R_{1,0}+R_{i}^{(\ddot{U},1,1)T}⊗MW_{1,0}+I_{n^{i}}⊗CW_{1,0}\right)\mathrm{vec}\left(R_{i,0}\right)+\\ \; & -\mathrm{vec}\left(M\Gamma_{i,0}^{\ddot{U}}+C\Gamma_{i,0}^{\dot{U}}+G_{2}\Gamma_{i,0}^{G_{2}}+G_{3}\Gamma_{i,0}^{G_{3}}\right). \end{array}$$

Note that the *i*-th linear system is under-determined in terms of the unknowns $W_{i,0}$ and $R_{i,0}$, as it consists of $\left(N\times n^{i}\right)+\left(n\times n^{i}\right)$ variables in $N\times n^{i}$ decoupled equations. When the left-hand side of Equation (Equation 24) is non-singular for every i≥2, a unique solution $W_{i,0}$ exists for any choice of $R_{i,0}$. The trivial choice that leads to a linear reduced velocity is $R_{i,0}=0$. When resonances cause the left-hand side of Equation (Equation 24) to become singular, two parametrization styles are possible for $R_{i,0}$: the normal form and the graph style ones. They are detailed in what follows.

#### 2.3.1. Normal Form Parametrization

The normal form method simplifies the reduced velocity by retaining only the components that lead to a singular left-hand side. The nonlinear terms in the reduced velocity are sought by first projecting Equation (Equation 24) onto its kernel and then by computing a partial inverse.

Said kernel is obtained by projecting the left-hand side of Equation (Equation 24) onto the subspace spanned by the master eigenmodes (see Equations (Equation 20) and (Equation 21)):(25)$${\left(e_{l}⊗\Phi_{j}\right)}^{T}\left[\left(R_{i}^{(\ddot{U},2,1)T}+2R_{i}^{(\ddot{U},2,2)T}\right)⊗M+R_{i}^{(\ddot{U},1,1)T}⊗C+I_{n^{i}}⊗K\right]\mathrm{vec}\left(W_{i,0}\right)=0,$$
(26)$$\left(\lambda_{l,i}^{\left(\ddot{U},2,1\right)}+2\lambda_{l,i}^{\left(\ddot{U},2,2\right)}\right)+2\chi_{j}\omega_{j}\lambda_{l,i}^{\left(\ddot{U},1,1\right)}+\omega_{j}^{2}=0,\;j\in \left[1,\;m\right],$$
where $\lambda_{l,i}^{\left(\ddot{U},1,1\right)}$, $\lambda_{l,i}^{\left(\ddot{U},2,1\right)}$, and $\lambda_{l,i}^{\left(\ddot{U},2,2\right)}$ are the *l*-th diagonal coefficients in $R_{i}^{\left(\ddot{U},1,1\right)}$, $R_{i}^{\left(\ddot{U},2,1\right)}$, and $R_{i}^{\left(\ddot{U},2,2\right)}$, respectively. Therefore, the kernel $N_{i}$ is the vector space of all elements $\left(e_{l}⊗\Phi_{j}\right)$ for which Equation (Equation 21) is fulfilled at some j∈[1,m]:(27)$$N_{i}=\mathrm{span}\left(\left(e_{l}⊗\Phi_{j}\right)\in R^{Nn^{i}}|\;j\in \left[1,\;m\right]\right).$$

Now, let $r_{i}$ be the number of structural resonances and $N_{i}\in R^{Nn^{i}\times r_{i}}$ be a basis for $N_{i}$, obtained by horizontally stacking the column vectors $\left(e_{l}⊗\Phi_{j}\right)$:(28)$$N_{i}=\mathrm{hcat}\left(N_{i}\right).$$

Then, the reduced velocity coefficients are determined by projecting the time-invariant Equation (Equation 24) onto $N_{i}$ to obtain:(29)$$\begin{array}{cc} \mathrm{vec}\left(R_{i,0}\right)= & -[N_{i}^{T}(I_{n^{i}}⊗MW_{1,0}R_{1,0}+R_{i}^{(\ddot{U},1,1)T}⊗MW_{1,0}+\\ \; & +I_{n^{i}}⊗CW_{1,0}){]}^{-1}N_{i}^{T}\mathrm{vec}\left(M\Gamma_{i,0}^{\ddot{U}}+C\Gamma_{i,0}^{\dot{U}}+G_{2}\Gamma_{i,0}^{G_{2}}+G_{3}\Gamma_{i,0}^{G_{3}}\right). \end{array}$$

#### 2.3.2. Graph Form Parametrization

Compared with the normal form parametrization, in this case, the kernel of the left-hand side contains all the eigenmodes of the master subspace. Hence, there is no need to determine the resonances, as they are automatically taken into consideration. Therefore, the reduced velocity is more complicated, since it also contains the non-essential terms (see [11]) and can be directly determined as:(30)$$\begin{array}{cc} \mathrm{vec}\left(R_{i,0}\right)= & -[{\left(I_{n^{i}}⊗\Phi \right)}^{T}(I_{n^{i}}⊗MW_{1,0}R_{1,0}+R_{i}^{(\ddot{U},1,1)T}⊗MW_{1,0}+\\ \; & +I_{n^{i}}⊗CW_{1,0}){]}^{-1}{\left(I_{n^{i}}⊗\Phi \right)}^{T}\mathrm{vec}\left(M\Gamma_{i,0}^{\ddot{U}}+C\Gamma_{i,0}^{\dot{U}}+G_{2}\Gamma_{i,0}^{G_{2}}+G_{3}\Gamma_{i,0}^{G_{3}}\right). \end{array}$$

Since the derivation of the reduced velocity through graph parametrization is similar to the normal form one, only the latter one is shown next.

### 2.4. Non-Autonomous Case

The previously discussed procedure for the autonomous case is now adopted to establish the nonlinear mapping and reduced velocity coefficients within a non-autonomous context. Starting with the leading order i=0, Equation (Equation 14) gives:(31)$$\begin{array}{cc} - & \Omega^{2}M\left(W_{0,1}^{+}e^{i\Omega t}+W_{0,1}^{-}e^{-i\Omega t}\right)+i\Omega C\left(W_{0,1}^{+}e^{i\Omega t}-W_{0,1}^{-}e^{-i\Omega t}\right)+K(W_{0,1}^{+}e^{i\Omega t}\;+\\ & +W_{0,1}^{-}e^{-i\Omega t})=-[M(i\Omega W_{1,0}\left(R_{0,1}^{+}e^{i\Omega t}-R_{0,1}^{-}e^{-i\Omega t}\right)+W_{1,0}R_{1,0}(R_{0,1}^{+}e^{i\Omega t}\;+\\ & +R_{0,1}^{-}e^{-i\Omega t}))+CW_{1,0}\left(R_{0,1}^{+}e^{i\Omega t}+R_{0,1}^{-}e^{-i\Omega t}\right)]+\frac{1}{2}F_{0}\left(e^{i\Omega t}+e^{-i\Omega t}\right). \end{array}$$

In the present non-autonomous case, the terms are further separated according to their harmonics. Specifically, the terms are collected in Equation (Equation 31) by way of $e^{i\Omega t}$ or $e^{-i\Omega t}$ according to:(32)$$\begin{array}{cc} \; & \left(-\Omega^{2}M+i\Omega C+K\right)W_{0,1}^{+}=-\left[MW_{1,0}\left(R_{1,0}+i\Omega I_{n}\right)+CW_{1,0}\right]R_{0,1}^{+}+\\ & -\left(M\Gamma_{0,1}^{\ddot{U},+}+C\Gamma_{0,1}^{\dot{U},+}+G_{2}\Gamma_{0,1}^{G_{2},+}+G_{3}\Gamma_{0,1}^{G_{3},+}\right)+\frac{1}{2}F_{0}, \end{array}$$
(33)$$\begin{array}{cc} \; & \left(-\Omega^{2}M-i\Omega C+K\right)W_{0,1}^{-}=-\left[MW_{1,0}\left(R_{1,0}-i\Omega I_{n}\right)+CW_{1,0}\right]R_{0,1}^{-}+\\ & -\left(M\Gamma_{0,1}^{\ddot{U},-}+C\Gamma_{0,1}^{\dot{U},-}+G_{2}\Gamma_{0,1}^{G_{2},-}+G_{3}\Gamma_{0,1}^{G_{3},-}\right)+\frac{1}{2}F_{0}. \end{array}$$

Equations (Equation 32) and (Equation 33) show that the solutions in terms of $W_{0,1}^{-}$ and $R_{0,1}^{-}$ are complex conjugates of $W_{0,1}^{+}$ and $R_{0,1}^{+}$, respectively. Thus, the derivation can be limited to the solutions given by $W_{0,1}^{+}$ and $R_{0,1}^{+}$ only.

In normal form parametrization, the kernel of the left-hand side of Equation (Equation 32) satisfies the following condition:(34)$$\begin{array}{c} -\Omega^{2}+2\chi_{j}\omega_{j}i\Omega +\omega_{j}^{2}=0,\;j\in \left[1,\;m\right]. \end{array}$$

Consequently, the reduced velocity is derived as:(35)$$\begin{array}{cc} R_{0,1}^{+}= & {\left\{N_{0}^{T}\left[MW_{1,0}\left(R_{1,0}+i\Omega \right)+CW_{1,0}\right]\right\}}^{-1}N_{0}^{T}[-(M\Gamma_{0,1}^{\ddot{U},+}+\\ & +C\Gamma_{0,1}^{\dot{U},+}+G_{2}\Gamma_{0,1}^{G_{2},+}+G_{3}\Gamma_{0,1}^{G_{3},+})+\frac{1}{2}F_{0}], \end{array}$$
where:(36)$$N_{0}=\mathrm{hcat}\left(N_{0}\right),\;N_{0}=\mathrm{span}\left(\left(\Phi_{j}\right)\in R^{N}|\;j\in \left[1,\;m\right]\right),$$
and $N_{0}\in R^{N\times r_{0}}$, with $r_{0}$ being the number of resonances.

For i=1, by collecting the terms associated with $e^{i\Omega t}$, Equation (Equation 14) leads to the following vectorized expression:(37)$$\begin{array}{cc} {}[ & \left(-\Omega^{2}I_{n}+2i\Omega R_{1}^{(\ddot{U},1,1)T}+R_{1}^{(\ddot{U},2,1)T}\right)⊗M+\left(i\Omega I_{n}+R_{1}^{(\ddot{U},1,1)T}\right)⊗C+\\ & +I_{n}⊗K]\mathrm{vec}\left(W_{1,1}^{+}\right)=-(I_{n}⊗MW_{1,0}R_{1,0}+i\Omega I_{n}⊗MW_{1,0}+R_{1}^{(\ddot{U},1,1)T}⊗MW_{1,0}+\\ & +I_{n}⊗CW_{1,0})\mathrm{vec}\left(R_{1,1}^{+}\right)-\mathrm{vec}\left(M\Gamma_{1,1}^{\ddot{U},+}+C\Gamma_{1,1}^{\dot{U},+}+G_{2}\Gamma_{1,1}^{G_{2},+}+G_{3}\Gamma_{1,1}^{G_{3},+}\right). \end{array}$$

The kernel of the left-hand side of Equation (Equation 37) fulfills the following conditions:(38)$$\left(-\Omega^{2}+2i\Omega \lambda_{l,1}^{\left(\ddot{U},1,1\right)}+\lambda_{l,1}^{\left(\ddot{U},2,1\right)}\right)+2\chi_{j}\omega_{j}\left(i\Omega +\lambda_{l,1}^{\left(\ddot{U},1,1\right)}\right)+\omega_{j}^{2}=0,\;j\in \left[1,\;m\right].$$

The reduced velocity is then obtained as:(39)$$\begin{array}{cc} \mathrm{vec}\left(R_{1,1}^{+}\right)= & -[N_{1}^{T}(I_{n}⊗MW_{1,0}R_{1,0}+i\Omega I_{n}⊗MW_{1,0}+R_{1}^{(\ddot{U},1,1)T}⊗MW_{1,0}+\\ & +I_{n}⊗CW_{1,0}){]}^{-1}N_{1}^{T}\mathrm{vec}\left(M\Gamma_{1,1}^{\ddot{U},+}+C\Gamma_{1,1}^{\dot{U},+}+G_{2}\Gamma_{1,1}^{G_{2},+}+G_{3}\Gamma_{1,1}^{G_{3},+}\right), \end{array}$$
where:(40)$$N_{1}=\mathrm{hcat}\left(N_{1}\right),\;N_{1}=\mathrm{span}\left(\left(e_{l}⊗\Phi_{j}\right)\in C^{Nn}|\;j\in \left[1,\;m\right]\right),$$
and $N_{1}\in C^{Nn\times r_{1}}$, with $r_{1}$ being the number of resonances.

For i≥2, by collecting the terms with $e^{i\Omega t}$, Equation (Equation 14) yields:(41)$$\begin{array}{cc} {}[ & \left(-\Omega^{2}I_{n}+2i\Omega R_{i}^{(\ddot{U},1,1)T}+R_{i}^{(\ddot{U},2,1)T}+2R_{i}^{(\ddot{U},2,2)T}\right)⊗M+\left(i\Omega I_{n}+R_{i}^{(\ddot{U},1,1)T}\right)⊗C+\\ & +I_{n}⊗K]\mathrm{vec}\left(W_{i,1}^{+}\right)=-(I_{n}⊗MW_{1,0}R_{1,0}+i\Omega I_{n}⊗MW_{1,0}+R_{i}^{(\ddot{U},1,1)T}⊗MW_{1,0}+\\ & +I_{n}⊗CW_{1,0})\mathrm{vec}\left(R_{i,1}^{+}\right)-\mathrm{vec}\left(M\Gamma_{i,1}^{\ddot{U},+}+C\Gamma_{i,1}^{\dot{U},+}+G_{2}\Gamma_{i,1}^{G_{2},+}+G_{3}\Gamma_{i,1}^{G_{3},+}\right). \end{array}$$

Now, the condition on the kernel is:(42)$$\left(-\Omega^{2}+2i\Omega \lambda_{l,i}^{\left(\ddot{U},1,1\right)}+\lambda_{l,i}^{\left(\ddot{U},2,1\right)}+2\lambda_{l,i}^{\left(\ddot{U},2,2\right)}\right)+2\chi_{j}\omega_{j}\left(i\Omega +\lambda_{l,i}^{\left(\ddot{U},1,1\right)}\right)+\omega_{j}^{2}=0.$$

The reduced velocity is obtained as:(43)$$\begin{array}{cc} \mathrm{vec}\left(R_{i,1}^{+}\right)= & -[N_{i}^{T}(I_{n}⊗MW_{1,0}R_{1,0}+i\Omega I_{n}⊗MW_{1,0}+R_{i}^{(\ddot{U},1,1)T}⊗MW_{1,0}+\\ & +I_{n}⊗CW_{1,0}){]}^{-1}N_{i}^{T}\mathrm{vec}\left(M\Gamma_{i,1}^{\ddot{U},+}+C\Gamma_{i,1}^{\dot{U},+}+G_{2}\Gamma_{i,1}^{G_{2},+}+G_{3}\Gamma_{i,1}^{G_{3},+}\right), \end{array}$$
where:(44)$$N_{i}=\mathrm{hcat}\left(N_{i}\right),\;N_{i}=\mathrm{span}\left(\left(e_{l}⊗\Phi_{j}\right)\in C^{Nn^{i}}|\;j\in \left[1,\;m\right]\right),$$
and $N_{i}\in C^{Nn^{i}\times r_{i}}$, with $r_{i}$ being the number of resonances at the *i*-th order.

### 2.5. Realification

As already pointed out above, the displacement vector is real-valued so that the complex coefficients obtained through the iterative procedure to solve Equation (Equation 12) must be transformed into real quantities. The transformation from complex to real values is performed as follows:(45)$$a=\left[\begin{array}{cc} I_{m} & I_{m}\\ -iI_{m} & iI_{m} \end{array}\right]p=\left[\begin{array}{cc} I_{m} & I_{m}\\ -iI_{m} & iI_{m} \end{array}\right]\left[\begin{array}{c} r\\ \bar{r} \end{array}\right]=Tp=2\left[\begin{array}{c} r^{ℜ}\\ r^{ℑ} \end{array}\right],$$
where $\bar{\left(•\right)}$ denotes the complex conjugate of •, and ${\left(•\right)}^{ℜ}$ and ${\left(•\right)}^{ℑ}$ denote the real and imaginary components, respectively. By means of this transformation and by way of Euler’s formula ($e^{ix}=\cos x+i\sin x$), one finally obtains:(46)$$\begin{array}{cc} U= & ℜ\left\{\sum_{i=1}^{q_{0}}W_{i,0}p^{⊗i}+\sum_{i=0}^{q_{1}}\left(W_{i,1}^{+}e^{i\Omega t}+W_{i,1}^{-}e^{-i\Omega t}\right)p^{⊗i}\varepsilon \right\}\\ = & \sum_{i=1}^{q_{0}}ℜ\left\{W_{i,0}{\left(T^{-1}\right)}^{⊗i}\right\}a^{⊗i}+\sum_{i=0}^{q_{1}}ℜ\left\{\left(W_{i,1}^{+}e^{i\Omega t}+W_{i,1}^{-}e^{-i\Omega t}\right){\left(T^{-1}\right)}^{⊗i}\right\}a^{⊗i}\varepsilon \\ = & \sum_{i=1}^{q_{0}}{\tilde{W}}_{i,0}^{ℜ}a^{⊗i}+\sum_{i=0}^{q_{1}}\left[\left({\tilde{W}}_{i,1}^{+,ℜ}+{\tilde{W}}_{i,1}^{-,ℜ}\right)\cos\left(\Omega t\right)-\left({\tilde{W}}_{i,1}^{+,ℑ}-{\tilde{W}}_{i,1}^{-,ℑ}\right)\sin\left(\Omega t\right)\right]a^{⊗i}\varepsilon . \end{array}$$

For what concerns the nonlinear reduced velocity coefficients, the time derivative of the realified reduced velocity must be also considered to obtain:(47)$$\dot{a}=\left[\begin{array}{cc} I_{m} & I_{m}\\ -iI_{m} & iI_{m} \end{array}\right]\dot{p}=T\dot{p}=2\left[\begin{array}{c} {\dot{r}}^{ℜ}\\ {\dot{r}}^{ℑ} \end{array}\right].$$

The nonlinear reduced velocity can be finally expressed as:(48)$$\begin{array}{cc} \dot{a}= & 2\sum_{i=1}^{q_{0}}\left[\begin{array}{c} {\tilde{R}}_{i,0}^{ℜ}\left(1\;:\;m,:\right)\\ {\tilde{R}}_{i,0}^{ℑ}\left(1\;:\;m,:\right) \end{array}\right]a^{⊗i}+2\varepsilon \sum_{i=0}^{q_{1}}\{\left[\begin{array}{c} {\tilde{R}}_{i,1}^{+,ℜ}\left(1\;:\;m,:\right)+{\tilde{R}}_{i,1}^{-,ℜ}\left(1\;:\;m,:\right)\\ {\tilde{R}}_{i,1}^{+,ℑ}\left(1\;:\;m,:\right)+{\tilde{R}}_{i,1}^{-,ℑ}\left(1\;:\;m,:\right) \end{array}\right]\cos\left(\Omega t\right)+\\ & +\left[\begin{array}{c} -{\tilde{R}}_{i,1}^{+,ℑ}\left(1\;:\;m,:\right)+{\tilde{R}}_{i,1}^{-,ℑ}\left(1\;:\;m,:\right)\\ {\tilde{R}}_{i,1}^{+,ℜ}\left(1\;:\;m,:\right)-{\tilde{R}}_{i,1}^{-,ℜ}\left(1\;:\;m,:\right) \end{array}\right]\sin\left(\Omega t\right)\}a^{⊗i}, \end{array}$$
where $\left(•\right)\left(1\;:\;m,:\right)$ is adopted to refer to the rows from 1 to *m* and all the columns of matrix •.

According to the procedure described in what precedes, the reduced-order model resulting from the Taylor-Fourier representation for mappings and reduced velocity has a dimension that depends on the number of master eigenmodes *m* retained in the model itself. In the absence of internal resonances between eigenmodes, a two-dimensional subspace is retrieved, regardless of the dimensionality of the original system [8]. The piezoelectric MEMS test structure studied in this work exhibits a resonance condition among the first three eigenmodes, thereby leading to a six-dimensional ROM. It should be noted that *m* can be determined in two ways, depending on the model being studied. If the spatial distribution of the external mechanical forcing is known in advance, *m* is derived by identifying when the eigenfrequencies are integer multiples of the excitation frequency or when resonance conditions (Equation 26), (Equation 34), (Equation 38), and (Equation 42) are satisfied. Alternatively, if the external mechanical forcing is unknown—such as when a structure is piezoelectrically actuated by an electric voltage, as in this study—a convergence analysis must be conducted to minimize the relative error between the frequency response curves (FRCs) from the reduced-order model (ROM) and full-order model (FOM) solutions. Further details are provided in Section 3.1.

Regarding the implementation of the DPIM method, the terms associated with the nonlinear internal forces, which are $\Gamma_{i,0}^{G_{2}}$, $\Gamma_{i,0}^{G_{3}}$, $\Gamma_{i,1}^{G_{2}}$, and $\Gamma_{i,1}^{G_{3}}$ in the homological equations, require specific computation that is detailed in Appendix B. Additionally, conditions (Equation 26), (Equation 34), (Equation 38), and (Equation 42) are considered fulfilled when the norm of their left-hand sides becomes smaller than a critical threshold, set to 0.01.

## 3. Numerical Results

The DPIM method described in Section 2 was adopted to compute the dynamic response of a polysilicon MEMS test structure designed with the goal of leading to stress intensification at the SiO_2_-polysilicon interface for mechanical characterization purposes. Since full-order models would be excessively time-demanding for the optimization of the shape of the moving structure to maximize the mentioned stress intensification, the model-order reduction strategy was implemented in MATLAB^®^ R2023a [14] and used in conjunction with COMSOL Multiphysics^®^ 6.2 [15] through the MEMS module and LiveLink^™^ for MATLAB^®^. The steps of the numerical procedure are illustrated in Figure 1 and can be briefly explained as follows. The nonlinear mapping and reduced velocity coefficients are first obtained, adopting a normal form parametrization to derive the reduced velocity coefficients (see [11]). The reduced-order dynamics is integrated in time by using an orthogonal collocation method with path length continuation, such as MATCONT [16]. The motion of the physical reference system is retrieved through Equation (Equation 11). Finally, the stress field is computed locally according to the constitutive law; in an FE analysis, this is carried out at the Gauss points.

The reference geometry of the test structure is shown in Figure 2, as designed in partnership with ST Microelectronics and Politecnico di Milano. The mechanical backbone is represented by a film of poly-crystalline silicon (termed *plate* in the figure), constrained at the two ends by massive anchors. The central portion of the plate is shaped to reduce its bending stiffness and to allow a flexible beam to be moved at one end in the out-of-plane direction (upward in the figure) whenever the plate is actuated. The PZT patches on top of the silicon layer serve to actuate (in the case of the larger ones) according to 3-1 mode piezo-mechanical coupling [1,17] or to sense the induced motion (in the case of the smaller ones).

As anticipated above, the goal of the present optimization procedure is the maximization of the stress field, specifically at the interface between the structural silicon layer and the silicon dioxide representing the substrate, to characterize the mechanical adhesion between the two. As reported next, the concept behind the design of this test structure is the maximization of the stress in the central portion of the device, where the flexible beam is anchored at one end, by maximizing the out-of-plane displacement at the other side of the same beam. A constraint to the optimization problem is represented by the strength of the silicon film: failure (if any) has to occur at the Si-SiO_2_ interface and not within the beam. Due to the statically indeterminate configuration of the plate, piezoelectric actuation induces bending and also an additional membrane-like stiffening effect (see [7,17]) that further complicate the analysis in relation to possible residual stress effects. Additional constraints to the optimization problem have to account for the microfabrication process and can be given in terms of relevant uncertainties (see, e.g., [18,19]), but also in terms of limited actuation voltage to avoid any breakdown of the PZT layer.

The electro-mechanical properties of the materials are reported in [7]. Rayleigh mass-proportional damping is assumed in the analysis, with a quality factor Q=100, which looks reasonable for MEMS devices operating at atmospheric pressure and in accordance with some preliminary tests carried out on the device. Accordingly, the coefficients in Equation (Equation 3) are set as follows:(49)$$\alpha =\frac{\omega_{1}}{Q},\;\beta =0$$
where $\omega_{1}$ is the natural frequency of the first eigenmode. In order to avoid building the ROM for the coupled electro-mechanical problem, the force induced by the electric potential applied to the PZT actuators is replaced by a body load, which has a spatial distribution proportional to the combination of the *m* master mechanical eigenmodes:(50)$$F\left(\Omega t\right)=V\sum_{i=1}^{m}\eta_{i}M\Phi_{i}\cos\left(\Omega t\right)$$
where $\Phi_{i}$ the *i*-th mechanical eigenmode and $\eta_{i}$ is a load multiplier, evaluated for each electrical eigenmode and determined by the surface charge collected at the top surface of the PZT layer where the electric potential *V* is applied. Given the symmetries in the geometry and actuation, only the first *m* symmetric modes are considered in the analysis. Therefore, only half of the device is discretized, as shown in Figure 3, to further reduce the computational burden, and 15 nodes, quadratic elements, are employed. The reported mesh was obtained as a result of a preliminary convergence analysis to assure it would be sufficiently fine to accurately capture the stress field in the region of interest.

The numerical investigation encompasses three steps: (1) Define the number *m* of master modes to retain in the ROM; (2) compute $q_{0}$ and $q_{1}$ (see Equations (Equation 11) and (Equation 12)) to ensure accuracy in the reconstructed displacement amplitude at point A (highlighted in green in Figure 2) induced by a forcing term able to trigger structural nonlinearities; (3) conduct the optimization of the device to maximize the stress intensification at varying beam length. All the analyses were run on a workstation equipped with an Intel(R) Xeon(R) W-2275 CPU featuring 14 cores, a processor frequency of 3.3 GHz, and 128 GB of RAM.

### 3.1. Setting the Number of Master Eigenmodes *m*

To prevent an excessive number of eigenmodes to be retained in the master subspace, namely, to avoid an excessively increasing computational cost to obtain the nonlinear mapping and reduced velocity coefficients, the effect of *m* is investigated first, keeping in mind that *m* must be high enough to capture potential internal resonances between the master modes. This goal is achieved by increasing *m* for low external forcing amplitudes to ensure the geometric nonlinearity remains inactive. For this reason, an electric potential of 0.1 V is adopted. Next, the frequency response curves obtained by using the ROM are compared with the corresponding FRC obtained with COMSOL Multiphysics^®^. Convergence is assumed to be achieved if the relative error between the FRCs from the ROM and the full-order model falls below the pre-defined tolerance.

Exemplary results, in terms of the first three linear eigenmodes, are displayed in Figure 4 as given by the FE solver. Although the shapes of the modes can slightly change at varying beam length, due to the different stiffnesses of the beam itself and of the plate in the reduced in-plane geometry central parts, they keep the same information, at least in terms of the effects on the out-of-plane motion of point A in Figure 3.

The results of the DPIM-based procedure relevant to the two beam lengths l=140 μm and l=220 μm are shown in Figure 5. In the charts, the FRCs are displayed at varying order *m* of the ROM and are compared to the FE solutions. The displacement $U_{zA}$ of point A is normalized by way of the silicon film thickness *h* to allow for the assessment of how good the DPIM-based solutions are in dealing with the geometrical nonlinearities. All in all, the plots demonstrate that relative to the FE results, the proposed solution converges for m=3, independently of the beam length. For this convergence analysis, $q_{0}=2$ and $q_{1}=0$ are specifically set.

### 3.2. Setting the Orders of the Polynomial Taylor Expansions

Having set the order *m* to attain the sought accuracy, the next step is the definition of the minimal Taylor series expansion orders $q_{0}$ and $q_{1}$ for the autonomous and non-autonomous solutions to ensure convergence in the structure response. Focusing again on the amplitude of the out-of-plane displacement $U_{zA}$ at point A, a voltage of 5 V is adopted to trigger the system nonlinearities. This voltage magnitude will also be used next in the geometry optimization process.

By adopting the same beam lengths examined in Section 3.1, the FRCs in Figure 6 are reported at varying orders $q_{0}$ and $q_{1}$ and compared with the FOM solutions. As shown in Figure 6c, the structure with l=220 μm exhibits a larger $U_{zA}/h$ ratio than the structure with l=140 μm (see Figure 6a), due to the higher compliance of the beam linked to its slenderness.

In the case l=140 μm, it is shown that convergence occurs for $q_{0}=3$ and $q_{1}=2$, with good agreement between the ROM and FOM curves. In these charts, the FOM results were obtained from a series of time-dependent solutions relevant to different excitation frequencies, with the amplitude of oscillations recovered after a sufficient number of cycles to assure that a steady state had been attained. Due to the higher displacement reached for the l=220 μm case, higher values of $q_{0}$ and $q_{1}$ are required. However, even for $q_{0}=5$ and $q_{1}=4$, the ROM struggles to accurately match the FOM results around the resonance frequency. This discrepancy can be minimized by constructing an ROM with a higher number of harmonics, as suggested in [20].

For the subsequent optimization process, $q_{0}=5$ and $q_{1}=4$ are then adopted in all the analyses at varying beam length, to keep a small error margin in the solutions.

A remark about the reported responses looks indeed necessary at this point. To experimentally obtain the curves shown in Figure 6c in the region characterized by multiple solutions (i.e., for $\Omega /\omega_{1}$ values between approximately 1.017 and 1.026), a frequency sweep is necessary: an upward sweep is required to capture the points in the upper part of the curve, while a downward sweep is necessary to obtain the points in the lower part. The solutions in the central portion of the curve (i.e., for $\Omega /\omega_{1}$ values between approximately 1.017 and 1.026 and for $U_{zA}/h$ values between approximately 0.36 and 0.75), commonly referred to as the unstable branch, can be just obtained numerically/theoretically and not experimentally.

### 3.3. Geometry Optimization of MEMS Testing Device

The final phase of the present analysis focuses on computing the FRC of the stress governing failure. Since the goal of the activity is the characterization of the structural interface between Si and SiO_2_, the maximum principal stress Sp is adopted as a reference. Even if not discussed here for brevity, two remarks must be made. First, the accuracy of the ROM in terms of the solution variables, like the considered displacement $U_{zA}$, and of space derivatives of the relevant field, like stress and strain components, is not assumed to be the same (or similar) at varying orders *m*, $q_{0}$, and $q_{1}$. At this stage, the convergence analysis reported above is considered to hold for different values of beam length *l*, in order to quantify the stress field at the SiO_2_-polysilicon interface. Second, a coarser mesh is employed in the convergence analyses in Section 3.1 and Section 3.2 compared with that shown in Figure 3. While the mesh in Figure 3 has approximately 600,000 degrees of freedom, the coarser mesh has around 110,000 degrees of freedom due to a reduced element density near the beam. This approach balances computational cost and accuracy in the displacement field as the primary variable for assessing convergence of the ROM in terms of the vertical displacement $U_{zA}$.

Figure 7 displays the domain where the stress field is computed, alongside the major findings in terms of the variation in Sp due to the beam length and excitation frequency. Structures with beam lengths between 60 μm and 160 μm look unsuitable, as the Sp in the polysilicon film exceeds that in the SiO_2_ layer for every value of the $\Omega /\omega_{1}$ ratio, suggesting that failure would likely occur in the polysilicon rather than at the SiO_2_-polysilicon interface. This outcome can be explained by considering that the contribution of the second mode progressively increases as *l* decreases, lowering the stress at the interface while increasing it at the rounded corners in the polysilicon region due to a torsional effect, as illustrated in the mode shape in Figure 4b. As the Sp in the SiO_2_ layer already attains high values at an applied electric potential of 5 V, only structures with beam lengths larger than 180 μm are considered, allowing for a sufficiently low electric potential for actuation and avoiding irreversible damage to the piezoelectric material.

A final comment is reported in relation to the computational costs of the presented results. Table 1 outlines the computation time for the different steps of the numerical procedure shown in Figure 1, for both the coarser and the finer meshes of the structure featuring l=220 μm. Additionally, the time required to retrieve the FOM time-dependent solutions for each excitation frequency in Figure 6 is also reported. The advantage of the DPIM method is evident, as the FOM solution for each excitation frequency and with the coarser mesh requires roughly eight times the time needed to set the ROM coefficients and the computation of the reduced-dynamics for a larger number of excitation frequencies (900 in Figure 6c). This computational efficiency stems as the main reason to avoid FOM analyses with the finer mesh, as the computational cost would become prohibitive. By using instead the DPIM approach, the full FRC for both displacement and maximum principal stress fields for each beam length is achieved in approximately 19 h.

## 4. Conclusions

In this work, a MEMS test structure with piezoelectric patches for actuation and sensing was presented. The results and the underlying methodology aimed at characterizing the polysilicon-silicon dioxide interface were discussed, given the potential for delamination due to stress concentration at the polysilicon-SiO_2_ interface. A reduced-order modeling approach, known as the direct parametrization of invariant manifolds (DPIM), was implemented in MATLAB R2023a^®^ and integrated with FE software COMSOL Multiphysics 6.2^®^ to optimize the geometry of the test structure and maximize stress intensification for mechanical characterization purposes.

The method to tune the parameters of the DPIM-based ROM was detailed. To achieve the sought accuracy in the maximum amplitude of the out-of-plane displacement of the plate to trigger dynamic nonlinearities and, therefore, attain stress intensification at the polysilicon-silicon dioxide interface that can lead to debonding under cyclic loadings, three eigenmodes were retained in the ROM. A Taylor-Fourier series expansion of order 5 for the autonomous cases and of order 4 for the non-autonomous cases was exploited in the DPIM. The frequency response function of the maximum principal stress at the interface was then studied at varying device geometry. The findings suggest that structures with beam lengths exceeding 180 μm yield stress at the polysilicon–SiO_2_ interface larger than within the polysilicon domain. This can be obtained by avoiding excessively high electric potentials, which could damage the piezoelectric layer. For shorter beam lengths, torsional effects induced by the second-mode shape would progressively reduce the stress at the interface while increasing it in the polysilicon region, thereby preventing the proper characterization of interface strength.

The validation of these findings will be reported through data to be collected in an experimental campaign. An enhanced ROM will be also implemented to reduce errors with respect to the full-order solutions, especially at large actuation voltages, by incorporating higher expansion orders for the forcing parameter and additional harmonics. The improved model will account for the residual stress, induced by the manufacturing process, and a possible static potential. Additional geometric parameters will be also explored, to achieve even larger concentrations of the stress field at the interface.

## Figures and Tables

**Figure 1 micromachines-15-01480-f001:**
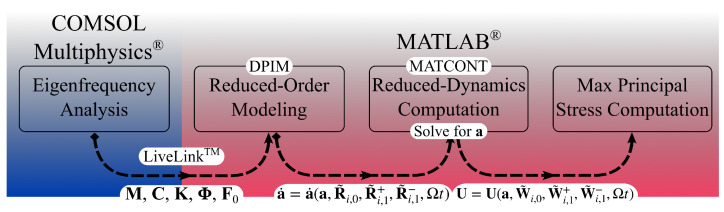
A sketch of the steps characterizing the proposed numerical procedure.

**Figure 2 micromachines-15-01480-f002:**
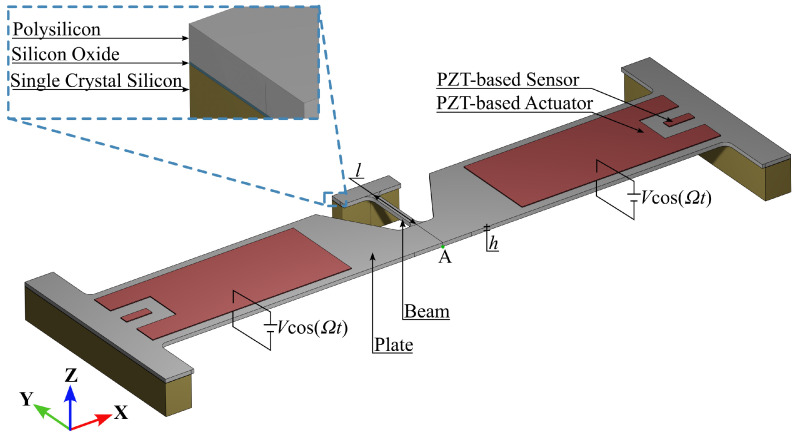
The geometry of the studied piezoelectric MEMS test structure and inset with details of the stacking sequence for the region under study.

**Figure 3 micromachines-15-01480-f003:**
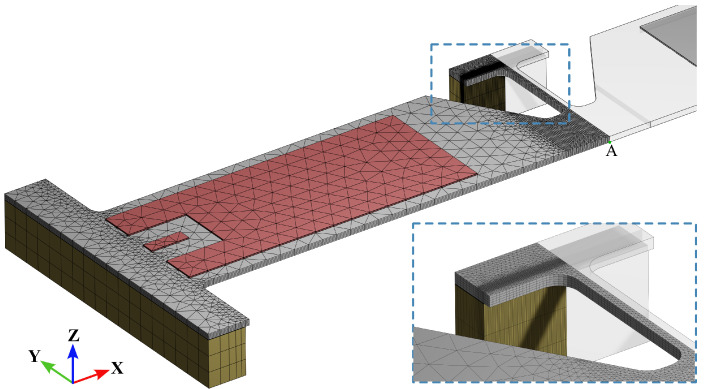
Finite element discretization of the test structure with inset to show a close-up of the region where the localized stress state is obtained with a more refined mesh.

**Figure 4 micromachines-15-01480-f004:**
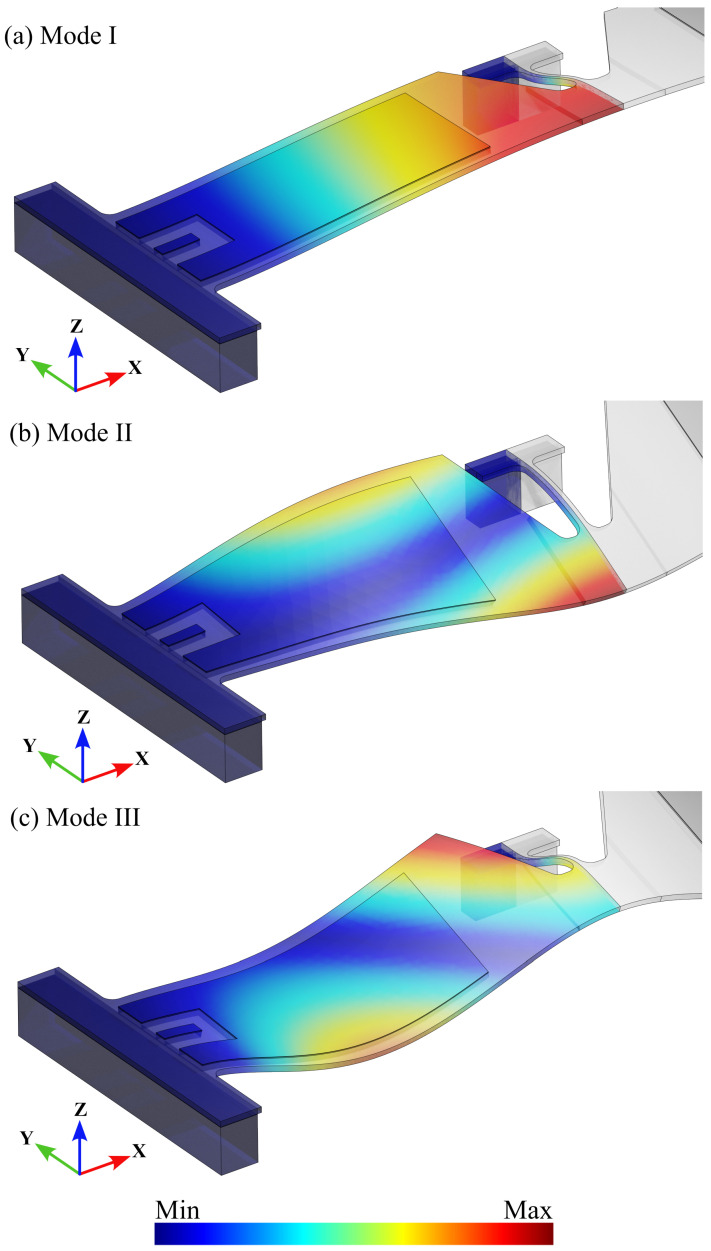
(**a**) First, (**b**) second, and (**c**) third fundamental eigenmodes of the test structure, plotted in terms of the amplitude of the mass-normalized displacement.

**Figure 5 micromachines-15-01480-f005:**
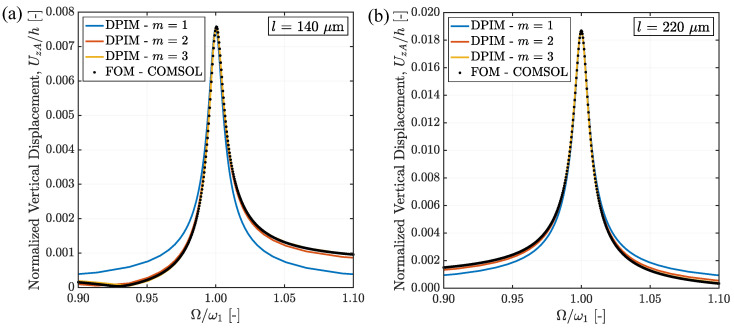
Comparisons the FRCs obtained with the FE analysis and the DPIM-based solution here proposed at varying order *m* of the ROM: (**a**) l=140 μm and (**b**) l=220 μm.

**Figure 6 micromachines-15-01480-f006:**
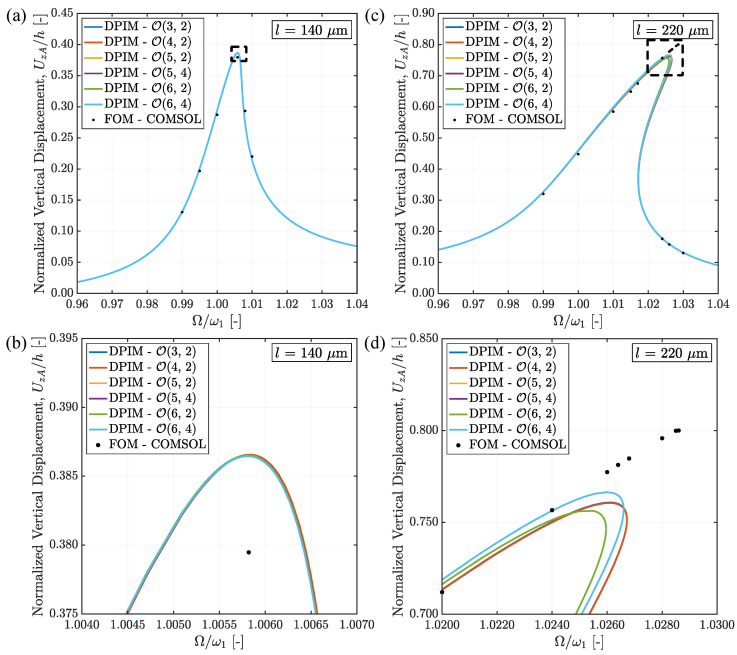
Comparisons between the FRCs obtained with the FE analysis and the DPIM-based solution at varying orders $q_{0}$ and $q_{1}$ of the ROM: (top) full FRCs and (bottom) close-ups near the resonance frequencies; (**a**,**b**) l=140 μm and (**c**,**d**) l=220 μm.

**Figure 7 micromachines-15-01480-f007:**
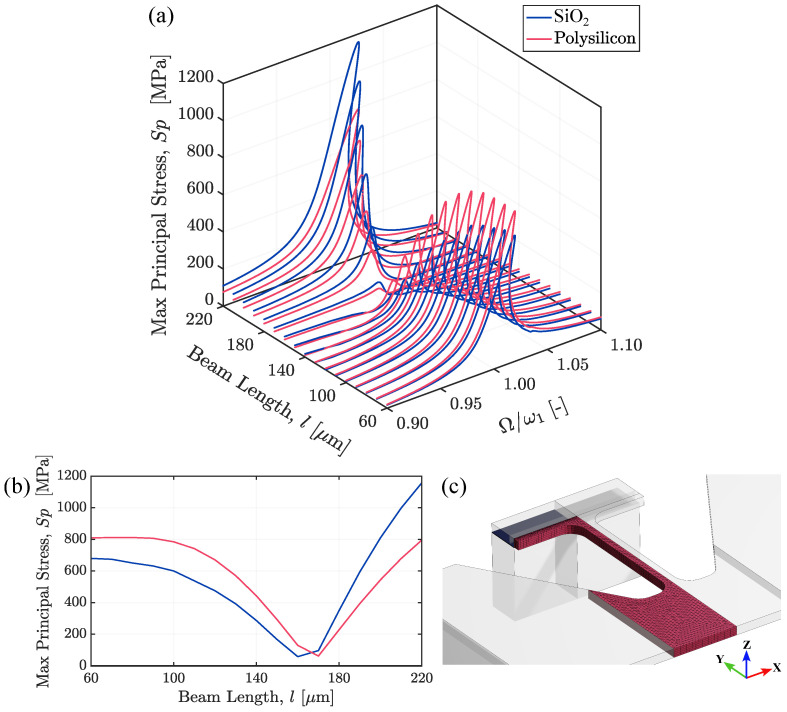
(**a**) Effects of beam length *l* and excitation frequency Ω on the maximum principal stress values in the polysilicon film and in the silicon dioxide layer. (**b**) The largest values of the maximum principal stress extracted at each beam length. (**c**) Domain where the stress has been computed, wherein the blue region marks the silicon dioxide area, while the red one is the polysilicon region.

**Table 1 micromachines-15-01480-t001:** Computation times for the test structure characterized by l=220 μm for the different steps of the reduced-order modeling procedure and for the full-order time-dependent solution.

Degrees of Freedom	Eigenfrequency Analysis	Reduced-Order Modeling	Reduced-Dynamics	Max Principal Stress	Full-Order Modeling
≈110,000	≈0.25 min	≈1.3 h	≈15 min	–	≈12 h
≈600,000	≈1 min	≈14 h	≈15 min	≈4.5 h	–

## Data Availability

The data presented in this study are available upon request from the corresponding author. The data are not publicly available due to some confidentiality issues.

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
