# Peer review of "Fatigue-Induced Failure of Polysilicon MEMS: Nonlinear Reduced-Order Modeling and Geometry Optimization of On-Chip Testing Device"

_micromachines, 2024, doi:10.3390/mi15121480_

Round 1

Reviewer 1 Report

Comments and Suggestions for Authors

The manuscript is interesting and well written.

Major comments

1) page 10, line 244: "The piezoelectric MEMS test structure...."

At this point of the treatment, the reader has no clue about the investigated MEMS structure. Maybe, the description of the geometry of the structure could be anticipated before the modeling? 

2) page 11, line 275: an explanation of the Gauss points would be beneficial for the reader

3) Figure 2: it would be helpful to see the inset as an expansion of the corresponding region of the structure. It is not clear from where the inset detail is taken. Also, specify in the caption that the detail is in the inset. 

4) page 13, line 336: point A is only in Figure 2 not 3. It would be useful to indicate point A also in Figure 3. 

5) I find quite odd to use omegaA lowercase to indicate the displacement (page 15, line 349 and figure 5). It confuses the reader. It would be much better to use zA, which would be in accordance with the 3D reference system in figure 4.

6) the effective Q value should be checked in the FRC in Figure 5. It looks like the Q is higher.

7) how, or why, did you choose the value of 5 V to induce nonlinear behavior?

8) Figure 6 is the most interesting to be expanded. The graphs are obtained from a series of time dependent solutions relevant to different excitations frequencies. Now, in 6c for the same frequency there are two possible values of displacement, which is typical of the nonlinear behavior of piezo actuated structures for large displacement. It is related to the fact that different displacements are induced for increasing frequency sweeps or for decreasing frequency sweeps. Would it be possible to specify theoretically how the curve is actually travelled for increasing and decreasing frequencies? That is actually what is then measured for example with an interferometer.

9) something similar holds for the graph is 7a. Would it be possible to predict what happens for increasing and decreasing frequencies? Actually, it could also happen that for decreasing frequencies the max stress remains lower.

Minor corrections:

Page 3, line 87: "...angular excitation frequency." instead of "...angular frequency excitation ."

Page 13, line 330: "Convergence is assumed to be achieved...", the "to be" is missing

Page 14, line 341: "All in all " requires a comma

Author Response

We would like to thank the reviewers for their careful reviews. We modified the text to reply to their comments. In addressing the comments and suggestions, main changes are highlighted in red in the new version of the manuscript.

Here below you will find our replies to all the comments.

Reviewer #1

The manuscript is interesting and well written.

We thank the reviewer for the positive comment.

  • page 10, line 244: "The piezoelectric MEMS test structure...."

At this point of the treatment, the reader has no clue about the investigated MEMS structure. Maybe, the description of the geometry of the structure could be anticipated before the modeling? 

The reference to the piezoelectric MEMS structure has been done in the end of Section 2 only to give an example of the dimensionality of the reduced order model and to show the way to retrieve the minimum number of master modes m. Up to this point of Section 2, no connection has been done to the actual piezoelectric MEMS test structure, because the aim is to give a general procedure to effectively create a reduced order model able to account for large transformations in mechanical systems, so with no limitation to piezo-electro-mechanical systems.

By keeping the geometry of the structure together with its finite element discretization in Section 3, the readers can focus more on the parameters characterizing the structure which are later used in the section to carry out the optimization procedure.

  • page 11, line 275: an explanation of the Gauss points would be beneficial for the reader

As requested, an explanation of the Gauss points has been given in this section.

  • Figure 2: it would be helpful to see the inset as an expansion of the corresponding region of the structure. It is not clear from where the inset detail is taken. Also, specify in the caption that the detail is in the inset.

As requested, the region of the structure corresponding to the inset has been better highlighted. Also, the caption has been changed accordingly.

  • page 13, line 336: point A is only in Figure 2 not 3. It would be useful to indicate point A also in Figure 3.

As requested, Figure 3 has been updated to include the point A.

  • I find quite odd to use omegaA lowercase to indicate the displacement (page 15, line 349 and figure 5). It confuses the reader. It would be much better to use zA, which would be in accordance with the 3D reference system in figure 4.

For better clarification, “wA” has been substituted with “UzA”, since “z” represents the material coordinate, instead of a component of the displacement vector.

It is also worth noting that the original notation was with the displacement identified as w (double-u), since displacements in the literature are often denoted as u,v,w along the three axes of an orthonormal reference frame. We know that omega is instead used for the angular frequency, but we do not have a control on the font adopted by the publisher.

  • the effective Q value should be checked in the FRC in Figure 5. It looks like the Q is higher.

We double checked and we rest assured that the effective Q value is correct. An electric potential of 0.1 V has been applied in this context “to ensure the geometric nonlinearity remains inactive”. The specific value of the adopted applied voltage has been added in this section.

  • how, or why, did you choose the value of 5 V to induce nonlinear behavior?

An electric potential of 5 V has been used to induce nonlinear behavior after analyzing the frequency response curve derived for the most compliant structure, which is characterized by a beam length of 220 µm. As evident in Figure 6c, a clear hardening behavior can be identified accompanied by a resonance frequency shift of about 2.5 %.

  • Figure 6 is the most interesting to be expanded. The graphs are obtained from a series of time dependent solutions relevant to different excitations frequencies. Now, in 6c for the same frequency there are two possible values of displacement, which is typical of the nonlinear behavior of piezo actuated structures for large displacement. It is related to the fact that different displacements are induced for increasing frequency sweeps or for decreasing frequency sweeps. Would it be possible to specify theoretically how the curve is actually travelled for increasing and decreasing frequencies? That is actually what is then measured for example with an interferometer.

Actually, the curve is just travelled for increasing frequencies. In the continuation algorithm, a starting frequency is chosen far away from the point in which there are two or three different values of displacement for the same frequency and then the solution relative to the first frequency point is determined. A continuation method similar to the pseudo-arclength continuation is then used to calculate the points along the path. The computation consists of the prediction of the next point and the correction of the predicted point. Given that the tolerance employed within the correction procedure is constant along the entire path, the curve obtained for decreasing frequencies should theoretically be identical to that for increasing frequencies.

  • something similar holds for the graph is 7a. Would it be possible to predict what happens for increasing and decreasing frequencies? Actually, it could also happen that for decreasing frequencies the max stress remains lower.

The maximum principal stress plotted in Figure 7a is actually determined after the time dependent reduced coordinates are calculated. Hence, as stated above and since the values of these variables do not change for increasing and decreasing frequencies, the stress does not change as well. This is due to the assumed (linear) elastic response of all the structural materials, for which a one-to-one relationship exists between deformation and stress measures.

Minor corrections:

Page 3, line 87: "...angular excitation frequency." instead of "...angular frequency excitation ."

Page 13, line 330: "Convergence is assumed to be achieved...", the "to be" is missing

Page 14, line 341: "All in all " requires a comma

Thanks for noting these typos. We have corrected the text as suggested.

Reviewer 2 Report

Comments and Suggestions for Authors

The research addresses a critical reliability issue in MEMS, namely fatigue-induced delamination, which is underexplored but highly relevant for industrial applications. Please find my comments to possible improve the paper:

1. Figures such as the finite element model and stress distribution maps (e.g., Fig. 4) are pivotal to understanding the results. However, they should include clearer labels and annotations to enhance clarity and interpretation.

2. The introduction is thorough but slightly dense. Simplifying the key takeaways and emphasizing the broader significance of the study would improve readability and engagement for the audience.

3. To be honest, some parts of Section 2, particularly in the methodological explanations, are overly technical. For this, it might benefit from simplification for broader accessibility.

4. Can experimental validation be added to support the computational results?

Author Response

We would like to thank the reviewers for their careful reviews. We modified the text to reply to their comments. In addressing the comments and suggestions, main changes are highlighted in red in the new version of the manuscript.

Here below you will find our replies to all the comments.

Reviewer #2

The research addresses a critical reliability issue in MEMS, namely fatigue-induced delamination, which is underexplored but highly relevant for industrial applications. Please find my comments to possible improve the paper:

  1. Figures such as the finite element model and stress distribution maps (e.g., Fig. 4) are pivotal to understanding the results. However, they should include clearer labels and annotations to enhance clarity and interpretation.

As requested, figures have been updated to improve the clarity and interpretation.

  1. The introduction is thorough but slightly dense. Simplifying the key takeaways and emphasizing the broader significance of the study would improve readability and engagement for the audience.

As requested, the introduction has been modified to improve the readability. A paragraph has been introduced at the end of the introduction, to highlight the key takeaways, as requested.

  1. To be honest, some parts of Section 2, particularly in the methodological explanations, are overly technical. For this, it might benefit from simplification for broader accessibility.

The approach shown in the present work is not available in the literature. For this reason, a simplified version of the full mathematical procedure has been reported in this paper, encompassing all the necessary instruments to replicate the findings of this research. A more simplified version may not be beneficial for readers who intend to apply the same approach on their future works.

  1. Can experimental validation be added to support the computational results?

To date, the experimental results are merely preliminary. We are planning to run them in the coming months and include the experimental validation in a follow-up of this research work.
